# Fabrication of PAN Electrospun Nanofibers Modified by Tannin for Effective Removal of Trace Cr(III) in Organic Complex from Wastewater

**DOI:** 10.3390/polym12010210

**Published:** 2020-01-15

**Authors:** Jing Zhang, Chao-Hua Xue, Hong-Rui Ma, Ya-Ru Ding, Shun-Tian Jia

**Affiliations:** 1College of Environmental Science and Engineering, Shaanxi University of Science and Technology, Xi’an 710021, China; zhangjing@sust.edu.cn (J.Z.); mahrxingfeng@163.com (H.-R.M.); 2College of Bioresources Chemical and Materials Engineering, Shaanxi University of Science and Technology, Xi’an 710021, China; dingyaru000@126.com (Y.-R.D.); jiashuntian@126.com (S.-T.J.); 3National Demonstration Center for Experimental Light Chemistry Engineering Education, Shaanxi University of Science and Technology, Xi’an 710021, China

**Keywords:** electrospun nanofibers, tannic acid, adsorption, trace Cr(III), complex form

## Abstract

Removal of chromium ions is significant due to their toxicity and harmfulness, however it is very difficult to remove trace Cr(III) complexed with organics because of their strong stability. Herein, a novel electrospun polyacrylonitrile (PAN) nanofibers (NF) adsorbent was fabricated and modified by tannic acid (TA) by a facile blend electrospinning approach for removal of trace Cr(III) in an organic complex. Utilizing the large specific area of nanofibers in the membrane and the good affinity of tannic acid on the nanofibers for hydrolyzed collagen by hydrophobic and hydrogen bonds, the as-prepared PAN–TA NFM exhibited good adsorption toward Cr(III)-collagen complexes and effective reduction of total organic carbon in tannage wastewater. The maximal adsorption capacity of Cr(III) is 79.48 mg g^−1^ which was obtained at the pH of 7.0 and initial Cr(III) concentration of 50 mg g^−1^. Importantly, the batch adsorption could decrease the Cr(III) concentration from 10–20 mg L^−1^ to under 1.5 mg L^−1^, which showed great application potential for the disposal of trace metal ions in organic complexes from wastewater.

## 1. Introduction

Chromium wastewater treatment is a focus of attention in industries such as dyeing and finishing, the electroplating industry, battery manufacturing and tanneries [1,2] because chromium ions in effluent can lead to severe environmental and health problems due to their non-biodegradability in living tissues [3,4,5]. Taking into consideration the toxicity of chromium, the maximum contaminant level (MCL) of total Cr in the discharge effluent is set at 1.5 mg L^−1^ in the leather industry of China. The main forms of chromium are Cr(VI) and Cr(III), which have different mobility and toxicity. Cr(III) ions are more stable than Cr(VI) in aqueous solution. Furthermore, trivalent Cr easily forms stable complexes with organic matter in water and oxidizes to the more toxic hexavalent form. Hence, the effective removal or at least a significant reduction of Cr(III) is critical, yet is also challenging, especially in the tanning industry [6].

Among traditional wastewater treatments, chemical precipitation is preferred because of its simple process and low cost. However, after the precipitation process, residual Cr of ~20 mg L^−1^ remains especially in the tannery effluent because of it also contains various organic pollutants. [7,8,9]. The existence of organic compounds such as hydrolyzed collagen in the tanning effluent changes the solubility of Cr(III) in aqueous solution and makes it more difficult to removal of Cr(III) from the wastewater by chemical precipitation [10]. Adsorption is the most suitable method for removal of heavy metals due to low cost, high efficiency, easy handling and lack of secondary pollution [3,9,11,12,13,14]. Therefore, recently much attention has been drawn to the application of nanofiber membrane materials for adsorption of inorganic chromium salts. Taha et al. [15] fabricated mesoporous polyvinyl pyrrolidone composite nanofiber membranes functionalized by amino (–NH_2_) to remove Cr(III) ions from wastewater by the adsorption method. Further, more than 97% of Cr(III) ions in the aqueous solution were removed by the nanofiber membranes. Chaúque et al. [16] prepared modified nanofibers (EDTA–EDA–PAN) to remove different ions including Cr(III) and Cd(II), and the modified membranes showed effective adsorption for both heavy metal ions; the maximum adsorption capacities were 32.68 and 66.24 mg g^−1^, respectively. Lou et al. [17] prepared and investigated the adsorption performance of the amino-functionalized polyacrylonitrile nanofiber membrane for Cd(II), Cr(III), and Cu(II) from aqueous solution, and the maximum adsorption capacities were 185, 204, and 105 mg g^−1^, respectively.

However, it should be noted that significant amounts of Cr(III) ions in tannery effluent are usually accompanied by a large concentration of organic substances. Cr(III) is easily chelated with organic matter forming different Cr(III)-bearing complexes [18,19]. Strong hydrophilicity and stability of organics, such as hydrolyzed collagen with carboxyl and amino groups in complex with Cr(III), have been supposed to be responsible for the residual Cr after wastewater treatment. It is very difficult to remove trace chromium from Cr(III) complexes in a highly dilute solution by traditional techniques including chemical precipitation, reverse osmosis, electrodeposition, or ion exchange [20].

Tannin (TA), a polyphenol macromolecule, is rich in phenolic hydroxyl groups and phenyl rings able to form a chelate with metal ions or protein [21,22,23], whereas hydrolyzed collagen consists of three polypeptide chains of triple-helical structures with both positive and negative charges and hydrophobic domains [24]. When mixing them, it is possible to interact polyphenols of tannic acid and hydrolyzed collagen, the mechanism for which has been proposed in the literature [25,26]. Given the synergistic and strong complexation interaction, hydrophobization of hydrolyzed collagen would occur, resulting in protein aggregation and precipitation. However, tannins dissolve easily in solution, so this method cannot be applied directly to remove heavy metals or protein in solution. To solve this problem, a lot of methods have been researched to immobilize tannic acid onto chitosan, cellulose, collagen and other water-insoluble substrates [27,28]. In this work, the partially hydrolyzed polyacrylonitrile was selected as the substrate. In recent years, polyacrylonitrile (PAN) which can be easily fabricated into nanofibrous materials by the electrospinning process, has been broadly applied for metal ion removal [15,16,29,30,31,32,33,34].

For the above reasons, in this study PAN nanofibers modified by tannins (PAN–TA) were prepared by electrospinning method. The electrospun PAN–TA nanofibers might be useful to absorb chromium ions, with their large surface area, and to fix the collagen complex with chromium by hydrophobic interactions, hydrogen bonds, and co-complexation to the tannins with a high content of multiple adjacent phenolic hydroxyls and phenyl rings, effectively reducing trace trivalent chromium in organic form in the effluent.

## 2. Experimental Section

### 2.1. Materials

Partially hydrolyzed polyacrylonitrile (PAN, 150 kDa molecular weight) was purchased from Shanghai Chemical Fibers Institute, Shanghai, China. Tannin powder and gelatin were supplied by Tianjin Tian Li Chemical Reagent Co., Ltd., Tianjin, China. *N*,*N*-Dimethyl formamide (DMF) was obtained from Jia Yu Chemical Corporation, Shanghai, China. The Cr(III)–collagen complex stock solution was prepared by dissolving appropriate amount of analytical grade Cr_2_(SO_4_)_3_·6H_2_O (Guangdong chemical engineering technology research and Development Center, Guangzhou, China) and gelatin in distilled water. The initial pH of solutions was adjusted by diluted HCl and NaOH solution. All solutions were prepared with deionized water.

### 2.2. Preparation of Electrospun PAN–TA Nanofibers

Briefly, the PAN–TA co-blended homogeneous solution was obtained by dissolving the tannic acid into DMF with magnetic stirring for 1.5 h at 298 K, followed by adding PAN powder under stirring for 6 h at 308 k. Different mass ratios of TA vs PAN (0:9, 1:9, 3:9, and 5:9) were adopted, and subsequently the nanofibers were marked as PAN, PAN–TA-1, PAN–TA-3, and PAN–TA-5. The spinning solution was loaded in a 10 mL syringe with a 22-gage needle. High voltage power (Dong Wen High Voltage Co., Ltd., Tianjin, China) was applied between the needle tip and a grounded collector wrapped in silicon paper. The electrospinning process was conducted using a high voltage of 20 kV, a tip-to-collector distance of 15 cm, and the flow rate of 0.5 mL h^−1^. Thus, the electrospun nanofibers were added into the glutaraldehyde to react at 50 °C for 4 h. Following that, the electrospun nanofibers were rinsed with deionized water and dried in a vacuum oven at 60 °C for 12 h. Finally, the crosslinked PAN–TA nanofibers (PAN–TA NFs) were obtained (Scheme 1).

### 2.3. Characterization

The surface morphologies of the nanofibers were observed by field emission scanning electron microscope (FE-SEM, Hitachi S-4800, Tokyo, Japan) operated at an acceleration voltage of 15 kV. The commercial software Nanomeasure1.2 was used to calculate the average diameter of the obtained electrospun nanofibers by measuring 100 different nanofibers. And the microstructure of nanofibers was further studied with transmission electron microscopy (TEM, FEI Tecnai G2 F20, Hillsboro, OR, USA). The chemical structures of different samples were performed on Fourier Transform Infrared spectrometer (ATR-FTIR, NICOLET iS10, Waltham, MA, USA). The spectral range was 4000–600 cm^−1^ with a resolution of 4 cm^−1^. Raman spectra was collected by a Thermo Fisher DXRxi spectrometer with a 532 nm laser at a power of 10 mw. X-ray diffraction (XRD) analysis was obtained using a Rigaku D/Max 2500 diffractometer with Cu Kα radiation (λ = 1.54 Å) (Rigaku D/Max 2500, Tokyo, Japan). The element composition of PAN and PAN–TA nanofibers were obtained using Energy Dispersive X-ray spectroscopy (EDX, EDAX Octane Elect Super, Philadelphia, PA, USA). The wettability of the nanofibers was studied via an optical contact angle system (OCA 20, Data Physics, Filderstadt, Germany). The energies and chemical state of different samples was analyzed by using X-ray photoelectron spectroscopy (XPS, ESCALAB 250XI, Thermal Fisher Corporation, Waltham, MA, USA). BET isotherms for surface properties of nanofibers were obtained by Micromeritics BET (ASAP 2020 M, Micromeritics Instruments Corporation, Norcross, GA, USA). The solution pH values were measured by using a pH meter (PHS-3C, Shanghai INESA Scientific Instrument Co., Ltd., Shanghai, China). The concentrations of Cr(III) ions in solution before and after adsorption were measured by an inductive coupled plasma emission spectrometer (ICP-OES, PerkinElmer ICP 2100, Waltham, MA, USA). The TOC content of the solution was analyzed by using the total organic carbon (TOC) analyzer (TOC-L CPN, Elementar Analysensysteme GmbH, Frankfurt, Germany).

### 2.4. Adsorption Studies of Electrospun PAN–TA Nanofiber Membrane

The obtained PAN–TA nanofibers were tested as adsorbents for removing of the Cr(III)–collagen complex from wastewater. All the batch adsorption experiments were performed using 50 mg of nanofibers, and 100 mL of Cr(III)–collagen solution, in an incubator shaker operated at 120 rpm (SHZ-82A, Changzhou Run Hua Electrical Appliances Co., Ltd., Changzhou, China).

#### 2.4.1. Influential Factors on Adsorption Capacity

In the adsorption process, the influence of important parameters including TA content (0–5 wt %), solution pH value (1–8), and contact time (10–240 min) on the adsorption of Cr(III) ions were researched by immersing 50 mg of nanofibers in Cr(III)–collagen solution in 250 mL flasks. The solution pH value was adjusted by using NaOH or HCl solution (0.1 mol L^−1^).

Firstly, the effect of different TA content of the nanofibers on the adsorption capacity was evaluated using a solution of Cr(III) at a concentration of 50 mg L^−1^ and hydrolyzed collagen (gelatin) at a concentration of 200 mg L^−1^, with shaking at 120 rpm for 4 h at 298 K.

Secondly, the effect of solution pH value on the adsorption capacity of was investigated. PAN–TA NFs were added into Cr(III)–collagen solution at pH levels from 1 to 8. After the mixture was shaken for 4 h, the PAN–TA NFs were pulled off the solution, and the concentration of Cr(III) and TOC content after adsorption were then measured. The data obtained at the optimum solution pH value was used to analyze kinetic curves for adsorption time from zero to 240 min, at 298 K, using an initial concentration of Cr(III) of 50 mg L^−1^. For the desorption–adsorption experiment, 50 mg PAN–TA nanofibers adsorbent was added into Cr(III)–collagen solution with shaking for 24 h at 298 K. The saturated PAN–TA NFs were regenerated using 0.1 mol L^−1^ HCI for 2 h and analyzed using ICP–OES and the TOC analyzer. After rinsing thoroughly with 100 mL deionized water, the membranes were recycled in adsorption–desorption process and the experiment was repeated for five cycles. The equilibrium adsorption capacity (*q*, mg g^−1^) of Cr(III) onto each PAN-based nanofiber membrane, the Cr(III) and TOC removal efficiency (%) of the samples was determined by following Equations (1)–(3):(1)$$q=\frac{V\left(C_{0}-C_{t}\right)}{m}$$
(2)$$R=\frac{C_{0}-C_{t}}{C_{0}}\times 100$$
(3)$$R_{TOC}=\frac{TOC_{0}-TOC_{t}}{TOC_{0}}\times 100$$
where *C*_0_ and *C_t_* are the initial and equilibrium concentrations of Cr(III) ions in the solution (mg L^−1^), respectively, *V* is the volume of Cr(III)–collagen solution (L), and *m* is the weight of the used PAN–TA NFs (g). *TOC*_0_ is the TOC value before adsorption by PAN–TA-3 NFs; *TOC_t_* is the TOC value after adsorption by PAN–TA-3 NFs.

#### 2.4.2. Kinetic Study

For demonstrating the improved adsorption capacity after PAN NFs were modified by TA, the adsorption kinetics onto both PAN NFs and PAN–TA NFs were measured. The two typical kinetic models (pseudo-first-order and pseudo-second-order) were applied to research the adsorption kinetics behavior of the nanofiber membrane. The linear expressions of two models are given by Equations (4) and (5):(4)$$\log(q_{e}-q_{t})=\log q_{e}-\frac{k_{1}t}{2.303}$$
(5)$$\frac{t}{q_{e}}=\frac{1}{k_{2}q_{e}^{2}}+\frac{t}{q_{e}}$$
where *q_e_* and *q_t_* (mg g^−1^) are the adsorption capacity at equilibrium time and time t, respectively. *k*_1_ (h^−1^) and *k*_2_ (g h^−1^ mg^−1^) are the pseudo-first-order and pseudo-second-order model rate constants, respectively.

#### 2.4.3. Adsorption Isotherm and Thermodynamics

The dynamic equilibrium process of adsorption process has often been predicted by using adsorption isotherm studies [35]. The Langmuir isotherm and Freundlich isotherm were applied for analyzing the results of the adsorption process of Cr(III) in complex form by PAN–TA nanofibers [36]. The expressions of two isotherm models can be expressed as Equations (6) and (7):(6)$$\frac{C_{e}}{q_{e}}=\frac{C_{e}}{q_{m}}+\frac{1}{bq_{m}}$$
(7)$$\log q_{e}=\log K_{F}+\frac{1}{n}\log C_{e}$$
where *C_e_* is the equilibrium concentration (mg L^−1^); *K_F_* and *n* are empirical constants that indicate the Freundlich constant and heterogeneity factor, respectively.

The adsorption isotherms were obtained by carrying out the typical adsorption experiments. with initial Cr(III) concentrations varying from 10 to 200 mg L^−1^, 50 mg of PAN–TA NFs was added to 100 mL Cr(III)–collagen solution at 298 k for 12 h. After adsorption, the solution was subsequently filtered through 0.45 μm membrane filter (Whatman), and the residual Cr(III) contents were analyzed by ICP-OES after the adsorption process.

The important thermodynamic parameters for removal of Cr(III)-collagen were explored considering the changes in the Gibbs free energy (∆*G*^0^, KJ mol^−1^), the enthalpy (∆*H*^0^, KJ mol^−1^), and the entropy (∆*S*^0^, KJ mol^−1^ K^−1^), described by the following Van’t Hoff Eqation (8)–(10) [37]:(8)$$lnK_{L}=-∆H^{0}/RT+∆S^{0}/R$$
(9)$$\Delta G^{0}=-RTlnK_{L}$$
(10)$$∆S^{0}=\left(∆H^{0}-∆G^{0}\right)/T$$
where *T* (K) is the absolute temperature, *R* (8.314 × 10^−3^ KJ mol^−1^ K^−1^) is the gas constant, and *K_L_* (L mol^−1^) is the Langmuir constant.

#### 2.4.4. Stability of the PAN–TA Nanofiber Adsorbent

The leaching of TA from the PAN–TA nanofiber adsorbent is an important factor in the adsorption process. The leaching mass of tannin from the adsorbent was investigated by the Prussian blue method [38]. This method is based on the reduction of ferric ion to ferrous ion in the presence of tannin, and then the formation of ferricyanide–ferrous ion complexes. The mixed solution has maximum absorbance at 706 nm. Therefore, the contents of tannins can be determined by testing the corresponding absorbance at 706 nm (A_706_) (as shown in Appendix A).

## 3. Results and Discussion

### 3.1. Preparation and Morphology

In this study, an effective adsorbent for trace Cr(III) in complex form was prepared by the co-blending electrospinning method. Thus, TA, which is a suitable candidate to adsorb heavy metal ions and hydrolyzed collagen protein, was chosen as the modified material.

SEM images, including raw PAN and PAN–TA nanofibers before crosslinking, are shown in Figure 1. It can be seen that the obtained nanofibers remain good fibrous morphology, which are continuous and random with few beads. The results indicated that the TA co-blending had little effect on the fibrous structures. Accordingly, with increasing tannic acid dosage, the average nanofiber diameter gradually increases from 150 to 230 nm. The morphology and size of the resultant fibers can be affected by the viscosity and surface tension of the electrospinning solution. In general, increasing the viscosity and surface tension favors the production of thicker fibers [39]. The larger nanofiber diameter is a result of the increased viscosity and surface tension of the electrospinning precursor solutions (as shown in Appendix A) [40].

EDX shows that the O content of the PAN was about 9.27%, while that of the PAN–TA-3 increased to 19.92% (Figure 2a,b). The C content of the PAN was about 54.94%, while that of the PAN–TA-3 increased to 55.14%. The N content of the PAN was about 35.19%, while that of the PAN–TA-3 decreased to 24.94%. These results imply that the PAN nanofibers were successfully modified by tannic acid with a high content of phenolic hydroxyls through facile co-blending electrospinning. As shown in Appendix A, the interior structure of the PAN and PAN–TA-3 nanofibers was further characterized by TEM. PAN–TA-3 nanofibers were obviously coated by a thin layer in comparison with PAN nanofibers. All the results indicated that PAN was successfully modified by tannic acid.

To further assess the modification of PAN by tannin, the wettability of the obtained nanofibers was analyzed (Figure 3). The original electrospun PAN nanofiber adsorbent has a contact angle of 48.5. After the co-blending modification, the contact angle of the nanofibers is decreased from 39.5 to 9.6 when the mass ratio of PAN to TA varies from 9:1 to 9:5, because of the high hydrophilicity of TA. The hydrophilicity of the adsorbent in aqueous solution is beneficial to the adsorption process.

### 3.2. Composition Characterization

To analyze the chemical structure of the different samples, ATR–FTIR spectra of PAN, PAN–TA NFs were obtained, shown in Figure 4a,b. The spectra exhibited absorption peaks at 2245 cm^−1^ attributed to the stretching vibrations of the nitrile group, and peaks at 1731 cm^−1^ (C=O), which suggested that the PAN was a copolymer of acrylonitrile and methylacrylate [41]. Meanwhile, the peaks at 3389 and 1660 cm^−1^ suggests the nitrile groups of polyacrylonitrile were partially hydrolyzed into amide groups [42]. These peaks were observed for all four nanofibers for the existence of PAN. The peak at 3303 cm^−1^ which was wide and strongly attributed to the stretching vibrations of the Ar–OH groups belonging to tannin [43] (Figure 4a). When tannins were added, especially when tannin content was greater than 3%, the peaks at 3389 cm^−1^ moved to 3370 and 3405 cm^−1^ and became wider and weaker which may be caused by the coincidence of phenolic hydroxyl and amino groups. When the tannin content increased from 3% to 5%, new peaks at 1608 and 1532 cm^−1^ appeared, which correspond to benzene ring vibrations [44]. Moreover, the intensities of peaks at 2245, 1731 and 1449 cm^−1^ decreased gradually with the amount of tannin increasing from 3% to 5% (Figure 4b). These results indicated that when the content of tannin was over 3%, PAN nanofibers were modified by tannin successfully. Raman spectra of PAN and PAN–TA-3 nanofibers showed that two new peaks at 1610 and 1713cm^−1^ of PAN–TA-3 nanofibers, attributed to the C=C bond belonging to the benzene ring, and the peak at 3066 cm^−1^ (Ar–OH), which suggested that the PAN was modified by TA successfully (Figure 4c). Further, the peak at 2242 cm^−1^ became weaker with increasing TA content. These results also indicated the modification of PAN nanofiber by tannic acid was successful.

XRD patterns of PAN and PAN–TA nanofibers were also studied and were shown in Figure 4d. Besides the tannin, all the patterns of the PAN and PAN–TA nanofibers show a similar diffraction peak at 2θ = 21.6° corresponding to the crystallographic plane (110) of PAN. During the procedure of electrospinning, the crystallization time of the polymer chain is shorter than required due to the simultaneous process of solvent evaporation and the solidification of the elongated nanofiber [45]. As Figure 4d shown, PAN–TA nanofibers show a similar amorphous peak of PAN at 21.6°, indicating that the PAN structure was well-preserved after co-blending with tannic acid.

A surface elements analysis was carried out on the PAN and PAN–TA NFs using XPS, as shown in Figure 5a–d. It was found that the oxygen content of the PAN modified by TA was four times higher than that of the original PAN. For the original PAN NFM, there are three different peaks at 283.95 eV of C–C/C–H bonds, 285.21 eV of C–O/C–N/C≡N bonds, and 286.75 eV of C=O bonds (Figure 5b). However, for PAN–TA NFs, the intensity of C-O/C-N/C≡N bonds was further strengthened due to the polyphenol groups which belong to tannic acid. For crosslinked PAN–TA NFs (Figure 5d), the peak at 286.7 eV of C–O and C=N bonds appeared [46], which is attributed to a crosslinking reaction between TA and polyacrylonitrile by glutaraldehyde. These phenomena further indicate that the PAN was successfully modified by tannin.

### 3.3. Batch Adsorption Experiments

The adsorption capacities of the as-prepared PAN–TA nanofibers toward trace Cr(III) in complex form were studied by the batch adsorption process. The co-blended ratio has a strong influence on adsorption capacity for the density of functional groups of the nanofiber membrane. As the results in Figure 6a show, the modified PAN–TA NFs adsorbent has relatively higher adsorption capacity toward Cr(III) in complex form than do raw (unlinked) PAN NFs. With the mass ratio of PAN vs TA gradually increased from 9:1 to 9:3, the adsorption capacity of nanofibers adsorbent increased quickly mainly due to the tannin content increasing, and the optimum adsorption capacity was 75.3 mg g^−1^, which had a removal efficiency was above 80%. This result indicated that the addition of tannin was beneficial to the adsorption of trace chromium in complexes. However, as the content of tannin increased, the BET surface area of nanofibers gradually decreased from 24.10 m^2^/g to 4.86 m^2^/g (as shown in Appendix A). The adsorption capacities of nanofibers adsorbents were affected by their surface area; accordingly when the content of the tannin was 5%, the adsorption of membrane decreased to some degree, even though the adsorption capacity of PAN–TA-5 nanofibers is 54.8 mg g^−1^, which is higher than PAN and PAN–TA-1 nanofibers. Therefore, for further research on the adsorption performance of nanofibers, PAN–TA-3 nanofiber adsorbent was used for removal of trace Cr(III) in organic forms for further research and given the abbreviation PAN–TA-3 NFs.

#### 3.3.1. Initial pH Effects

The effect of solution pH value on removal of Cr(III)–collagen complexes by PAN–TA-3 NFs was investigated in a pH range from 1.0 to 8.0. As shown in Figure 6b, the solution’s initial pH had a great influence on removal of Cr(III). At the beginning, the adsorption capacity of PAN–TA-3 NFs to Cr(III) slightly increased in the pH range from 1.0 to 2.0, then significantly increased in the pH range from 2.0 to 7.0, and then decreased slightly in the pH range of 7.0–8.0. These results indicated that the adsorption of Cr(III) onto PAN–TA-3 NFs was highly pH-dependent with a maximum adsorption capacity of 79.48 mg g^−1^ at pH 7.0. In order to confirm that Cr(III) is removed in the form of an organic complexation, the TOC content of the solution after adsorption under different pH was also investigated (Figure 6b). The results showed that the TOC value decreased sharply in the range from 2.0 to 4.0, decreased slightly in the pH range of 4.0–7.0, and then increased slowly in the pH range of 7.0–8.0. The *R_TOC_* of the solution is about 84.5% at pH 6.0–7.0. This indicated that the Cr(III) ions and hydrolyzed collagen were removed simultaneously by PAN–TA NFs from the solution. Thus, for reducing the concentration of Cr(III) ions to a greater extent from the wastewater, the ideal solution initial pH value is 7.0.

#### 3.3.2. Kinetic Study

In order to demonstrate the improved adsorption capacity of PAN nanofibers after modification by tannic acid, the adsorption kinetics and capacities of the PAN–TA-3 and PAN NFs were researched. Two typical kinetic models were used to analyze the adsorption kinetics behavior of the adsorbents (Figure 6c, Appendix A). The adsorption process of the PAN–TA-3 NFs was divided into two periods, a rapid adsorption at the initial stage, followed by a relatively slow stage until the adsorption equilibrium was reached (4 h). The linear fitting results are shown in Table 1. According to the correlation coefficient (*R*^2^), the adsorption data of the PAN–TA-3 NFs are better fitted with the pseudo-second-order model (the inset in Figure 6c is their pseudo-second-order kinetic plots). In order to identify the adsorption capacity of the PAN–TA-3 NFs, the surface morphologies of PAN and PAN–TA-3 NFs after adsorption of Cr(III)-collagen complexes for over 120 min have been observed by SEM (Figure 6e,f). It can be seen that more adsorbates are present on the surface of the modified nanofibers than on raw PAN nanofibers. All these results show that PAN–TA-3 NFs have a much better adsorption capacity toward trace Cr(III) in complexes than PAN NFs.

#### 3.3.3. Adsorption Isotherms and Thermodynamics

First, the adsorption isotherms were studied by two classical isotherm models (Figure 6d and Appendix A), and linear fitting results are shown in Table 2. The results illustrate that the adsorption data of Cr(III) onto PAN–TA-3 NFs are better fitted with the Langmuir isotherm (Figure 6d), with an *R*^2^ value beyond 0.99, which reflects that the adsorption is a monolayer adsorption process and the adsorption sites on the absorbent are homogeneous. In addition, the maximum adsorption capacity (*q_max_*) from the Langmuir fitting for Cr(III) adsorption onto PAN–TA-3 NFs is 147.06 mg g^−1^, which agrees with the experimental data.

Secondly, the plots of ln K_a_ versus 1/T are shown in Appendix A, and the thermodynamic parameters for adsorption of trace Cr(III) in complex using PAN–TA-3 NFs are provided in Appendix A. The calculated *ΔS*^0^ is 0.115 (KJ mol^−1^), and its positive value corresponds to increasing randomness at the solid–liquid interface through the adsorption of Cr(III) onto PAN–TA-3. The positive ∆*S*^0^ suggests that the entropy change caused by desorption is greater than that caused by adsorption of the Cr(III)–collagen complex. Further, the calculated ∆*G*^0^ at 298, 308, and 318 K are −5.088, −6.346, and −7.388 (kJ mol^−1^), respectively. The negative ∆*G*^0^ means that the adsorption process is spontaneous and feasible. The increasing of temperature is beneficial to the adsorption of the PAN–TA-3 NFs to the Cr(III)–collagen complexes. Finally, the calculated ∆*H*^0^ (28.076 kJ mol^−1^) confirmed that increasing adsorption temperature benefits the adsorption process and is endothermic.

#### 3.3.4. Trace Cr(III) in Complex Removal

As described in previous literature [6], it is a challenge to decrease the concentration of trace Cr(III) in complex form to below 1.5 mg L^−1^, which is important for real-world, environmental water-treatment applications, especially in the leather factory [47]. To investigate the ability of PAN–TA-3 NFs to remove trace Cr(III) in complex form, the effects of adsorbent dosage of PAN–TA-3 NFs and initial concentration of Cr(III) on the residual Cr(III) concentration in solution were analyzed. As Figure 7a shows, when the dosage of absorbent was 1.0 or 2.0 mg mL^−1^, the Cr(III) concentration decreased from 10 mg L^−1^ to below 1.5 mg L^−1^ after the adsorption process. When the initial Cr(III) concentration of 20 mg L^−1^, the dosage of adsorbent was 2.0 mg mL^−1^, the residual Cr(III) concentration fell below 1.5 mg L^−1^ (Figure 7b). The results reveal that PAN–TA-3 has a potential application to reduce the concentration of trace Cr(III) in complex to under 1.5 mg L^−1^ (MCL), which is an ideal method for tanning wastewater.

#### 3.3.5. Adsorption Cycles

For the better practical applications, the reuse of PAN–TA-3 NFs was investigated in this study. After each adsorption–desorption cycle, the PAN–TA-3 NFs were soaked in distilled water and dried at room temperature. The removal efficiency of Cr(III) and TOC still remained above 80% and 70% after five cycles of adsorption–desorption experiments (Figure 7c). The SEM image of PAN–TA-3 NFs after the five cycles showed that the nanofibers still retained good fibrous morphology (Appendix A), indicating that PAN–TA-3 can be regenerated and reused under the experimental conditions. Furthermore, Figure 7d,f show that the solution become very transparent after adsorption by PAN–TA-3 NFs. Thus, the results imply that PAN modified by TA benefits the adsorption process.

A comparison of the adsorption capacity of PAN–TA-3 NFs to some other tannin-based adsorbers is presented in Table 3 [1,46,48,49,50]. It can be seen that PAN–TA-3 nanofiber adsorbent has a relatively high adsorption capacity for Cr(III) compared to other tannin-based adsorbents, while lower than tannin-based adsorbents for Cr(VI). Moreover, PAN–TA-3 nanofibers can remove the trace Cr(III) and hydrolyzed collagen in the solution at the same time, which suggested that PAN–TA-3 nanofiber adsorbent is an efficient material for the removal of trace Cr(III) in complexes from aqueous solutions.

### 3.4. Stability of PAN–TA-3 Nanofiber Adsorbent

Considering practical applications, the PAN–TA-3 nanofiber adsorbent should remain stable during the adsorption process. Based on the Prussian Blue analysis method of tannin content, the concentration of tannin leaked into Cr(III) solution after adsorption was found to be 3.95 mg L^−1^, which indicates that the stability of PAN–TA-3 nanofibers is satisfactory during the adsorption process.

### 3.5. Adsorption Mechanism

Investigation of the element distribution by EDS of the PAN–TA-3 nanofibers before and after adsorption showed that the dominant elements (C, O, N) were uniformly distributed on the PAN–TA-3 nanofiber before and after adsorption (Appendix A). The Cr content of the PAN–TA-3 after adsorption was about 21.12%, while the S content of the PAN–TA-3 after adsorption was about 1.96%, which belonged to gelatin (Appendix A), implying the successful adsorption of Cr(III)–collagen complexes onto the PAN–TA-3 nanofibers.

For further analyzing the adsorption mechanisms, an XPS analysis was applied to explore the energies and chemical state of PAN–TA-3 NFs before and after adsorption of Cr(III) in complex form. In the XPS spectra of PAN–TA-3 after Cr(III)–collagen adsorption (Figure 8a), two new peaks could be seen between 571.77 and 593.17 eV, assigned to the Cr_2p_ region [51]. The analysis confirmed the successful adsorption of Cr(III) onto PAN–TA-3 NFs. In the meantime, the intensity of N_1s_ showed a conspicuous increase (Figure 8b). The result indicated that the density of N_1s_ increased because of the hydrolyzed collagen absorbed onto the PAN–TA-3 nanofibers during the adsorption process. After adsorption, the binding energy of O_1s_ shifted to a lower value, with the peak width (W) of O_1s_ changed to 9.7 from 8.2 eV (Figure 8c). The results suggested that the O participated in the adsorption process of Cr(III)–collagen complexes.

After adsorption of Cr(III)–collagen complexes by the PAN–TA-3 nanofibers, two peaks of Cr_2p1/2_ and Cr_2p3/2_, which could be further split into two additional peaks are shown in Figure 8d [52,53]. The peaks at 577.02 eV (Cr_2p3/2_) and 586.77 eV (Cr_2p1/2_) are attributed to Cr(III), and the peaks at 578.25 eV (Cr_2p3/2_) and 585.62 eV (Cr_2p1/2_) are attributed to Cr–O bonds, suggesting that Cr(III) and Cr–O co-exist on the nanofiber surfaces. Because the polyphenol group of the tannin was the only functional group, this indicates that Cr(III) ions and Cr(III)–collagen complexes were absorbed simultaneously by polyphenolic groups of the PAN–TA-3 NFs.

## 4. Conclusions

In this study, PAN–TA nanofibers were fabricated successfully through a facile co-blended electrospinning method for removal of trace Cr(III) in organic complexes, especially those found in tanning wastewater. Due to the plentiful polyphenol groups of the tannin, it was found that the nanofiber adsorbent had a good adsorption capacity for the removal of trace Cr(III) in complex form compared with other adsorbents. The maximal adsorption capacity of Cr(III) onto PAN–TA nanofibers is 79.48 mg g^−1^ and the adsorbent showed a good recycling ability. The adsorption behaviors were found to describable by the pseudo-second order model and Langmuir model very well. Meanwhile, the hydrolyzed collagen, which is stable in water, could be removed by PAN TA to a great extent. Moreover, the Cr(III)-bearing complexes, which are difficult to remove by common chemical precipitation, could easily be absorbed onto the PAN TA nanofibers. Therefore, PAN TA nanofiber adsorbent is an ideal candidate for effective removal of trace Cr(III) in complex from aqueous solutions, especially Cr(III)-collagen in tanning wastewater. This strategy of modification of absorbent by tannic acid could pave the way to an effective removal of trace metal ions with stable organic pollutants from wastewater in practical application.

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
