# Peer review of "Fabrication of PAN Electrospun Nanofibers Modified by Tannin for Effective Removal of Trace Cr(III) in Organic Complex from Wastewater"

_polymers, 2020, doi:10.3390/polym12010210_

Round 1
Reviewer 1 Report
As chromium wastewater treatment is actual problem today the contents presented in paper “Fabrication of PAN Nanofiber Membranes Modified by Tannin for Effective Removal of Trace Cr(III) in Organic Complex from Wastewater” should be attractive to the readers of the Polymers journal. This pollution process is very actual problem today worldwide in different areas.
The paper is easy to read, the experimental procedure is satisfactory, and the synthesized PAN-TA nanofiber membrane are well characterized and the applied.
I can recommend publication of the following manuscript with minor revision.
The following are some comments:
Experimental section
It is recommended that Scheme 1 should be moved to the experimental part into a section 2.2.
Results and discussionsIt should be pointed why PAN-TA 3 stood out as the best for further analysis for purposes of removal of trace Cr(III)in complex from aqueous solutions.
Conclusions
Please, rewrite Conclusion. It is necessary to highlight main findings, such as: why PAN-TA 3 is better than other materials, as well as the percentage of chromium removal relative to others etc.
ReferencesReference list is up to date and more than adequate. However, PAN is commonly used for the preparation of recyclable mats using the electrospinning technique. Please use some more corresponding references on it.
Reviewer 2 Report
The authors claimed that the developed membrane is able to treat remove Cr(III) from the effluent discharged by leather industry. But, authors only prepared a very simple solution without considering the presence of many other organic pollutants in tannery effluent. Authors should study the effects of some pollutants on the performance of membranes. Besides, the leaching of TA from PAN nanofiber should be studied as it is important criteria for practical use.
Another main concern is the use of “membrane” in this work. The work has nothing to do with filtration as the typical membrane process (pressurized within a permeation cell) is not employed. Author is advised to amend the term accordingly!
Abstract
Authors should highlight the results in absorption capacity (mg/g) rather than reduction of Cr(III) concentration level.
Section 2.2
Authors can’t use “%” for mass ratio. Ratio is dimensionless!
Scheme 1 should be presented in Section 2 instead of Section 3 because it is about the approach!
Section 3
Figure 1 - Detailed discussion is missing. What is the effect of TA in reducing the diameter of fiber from 150 – 230 nm?
Figure 3 – Author claimed that new peak at 3370 cm-1 is due to the Ar-OH of TA. But, other samples also show a broad peak in the region of 3250 – 3500 cm-1! The results of membrane surface contact angle should be presented in main text rather than as supplementary.
Figure 5a – Detailed discussion is missing. What are the main reasons that the q is significantly increased from PAN-TA-1 to PAN-TA-3 and then decreased in the case of PAN-TA-5? Discussion should be provided with strong supporting evidence.
Figure 5e and f – Both images are not discussed in the text!
A table to compare the performance of the developed nanofiber membranes with other similar studies should be provided followed by in-depth discussion.
Lastly, English should be checked carefully. For instance, “has been draw” should be “has been drawn”. For the reference cited in the text, it is only the FIRST name of author should be mentioned!
Round 2
Reviewer 2 Report
I'm now satisfied with the correction/changes made by the authors. This paper can now be accepted for publication.